# Knowledge Mapping Analysis of the Study of Rural Landscape Ecosystem Services

Yinyi Wang [1,†], Yaping Zhang [1,†], Guofu Yang [2], Xiaomeng Cheng [1], Jing Wang [1] and Bin Xu [1,*]

[1]  School of Landscape Architecture, Zhejiang A&F University, Hangzhou 311300, China
[2]  School of Art and Archaeology, Zhejiang University City College, Hangzhou 310015, China
*   Correspondence: 20010051@zafu.edu.cn; Tel.: +86-138-1919-5039
†   These authors contributed equally to this work.

**Abstract:** Understanding the research lineage of rural landscape ecosystem services (RLESs) is of importance for improving rural landscapes and developing sustainable ecosystem services. However, there is currently no literature analysis on the scientific quantification and visualization of RLESs. In this study, 4524 articles related to RLESs from 1990 to 2021 were analyzed using the bibliometric method and ISI Web of Science database. The results show that RLES research hotspots have gradually shifted from the early keywords of "vegetation", "land use change", "agriculture", "rural gradients" and "models" to the emerging "cultural ecosystem services", "rural tourism", "landscape preferences" and "policy guidance". Scholars from developed and developing countries place different emphases on research hotspots in terms of research content, scale and methodology due to differences in their research backgrounds and other aspects. In addition, five categories of research fronts were obtained through literature co-citation analysis. Through burst word detection analysis, combined with basic research and research hotspots and frontier analysis, we concluded that future RLES research will focus on four areas: (1) the relationship and collaboration between and management of biodiversity and ecosystem services; (2) the landscape value of RLESs; (3) land-use changes and ecosystem service values; and (4) research methods for innovative RLESs. Our findings may contribute to better in-depth RLES research by providing a theoretical reference and practical help for future related research.

**Keywords:** rural ecosystems; biodiversity; bibliometrics; visualization analysis; research frontiers and trends; literature review

## 1. Introduction

Ecosystem services (ESs) refer to all ecosystem goods and functions that support and satisfy human survival and development. They encompass the various benefits that humans obtain from ecosystems, including both tangible material goods and intangible services. The latter are divided into four types: supply services, regulating services, cultural services and support services (those necessary to maintain other types of services). Together, they form a basis for the harmonious coexistence of humans and nature [1–3]. In the 1970s, the United Nations University (UNU) published the Report on Human Impact on the Global Environment in 1970, which first introduced the concept of ESs and listed the environmental service functions that ecosystems provide to humans [4]. Later, Holder and Ehrlich (1974), Westman (1977) and Odum (1986) successively conducted early and influential studies and successively conducted studies on global environmental services functions and natural services functions, pointing out that the loss of biodiversity had a direct impact on ecosystem services and thus generating the concept of ES functions [5–7]. Early ES research focused on the introduction of concepts and the construction of a theoretical framework. Today, the concept is increasingly used to inform land-use planning, economic decision-making and biodiversity conservation [8–11]. Because of its immediate relevance to human well-being, ES research has received significant attention from scholars and research organizations across various sectors [12–15].

The provision of and demand of ESs exhibit strong spatial-scale characteristics [16,17]. Since the beginning of the 21st century, scholars have carried out many practical studies on global and regional scales, watershed scales, single ecosystem scales and single service values. In addition, previous studies have focused on large- or medium-scale areas, such as watersheds and city–county areas. However, studies focusing on rural areas have been limited [18–21].

Rural ecosystems are currently some of the most severely affected by humans in the world, and rural landscapes have undergone profound changes due to the effects of rapid urbanization. As a comprehensive and complex research field, RLESs have received increasing attention from researchers. There are many related studies and papers, but no scholars have reviewed and summarized the literature in this field or qualitatively defined the concept of RLESs. Based on previous research, this paper proposes RLESs as research on rural ecosystem services from the perspective of landscape, which is the service function generated by the interconnection between landscape elements and natural ecological processes and has comprehensive ecosystem service functions, including ecological, economic, aesthetic, cultural and other values [22–24]. Studies of RLESs can reflect the overall status of rural ecosystems, reveal the contribution of ecosystems to human well-being and evaluate the supporting role of ecosystems in economic and social development and the ecological relationships among regions. However, the starting point of many current studies is to take the rural ecosystem as an important support system for the urban ecosystem or to carry out research from the perspective of the environment and resources. Few scholars have studied the dynamic evolution process, characteristics and existing problems of modern RLESs from the perspective of the rural ecosystem itself. Therefore, understanding the development of RLESs and proposing areas for future research are crucial for improving rural landscapes and creating resilient rural ecosystems [25–27].

With the rapid development of bibliometrics, it has become a research hotspot in various academic fields to provide new perspectives for the knowledge structure and development of a scientific field through data mining, information processing and visualization techniques. Many researchers use bibliometric methods to provide innovative perspectives on evaluating research trends [28]. Statistical analysis (including publication volume, journals, countries and institutions) can help people interested in a field to quickly grasp the basic information and development of the literature. Co-citation analyses enable researchers to recognize inherent relationships in the literature and identify key knowledge groups in core publications/citations and fields [29]. In addition, mapping and visual bibliometrics can illustrate the relationships between units of analysis in a more intuitive way.

Over the past decade or more, there has been a considerable amount of quantitative research in the literature on various topics related to ESs, including urban ecosystem services (UESs) [30], agricultural ecosystem services (AESs) [31], mountain ecosystem services (MESs) [32], forest ecosystem services (FESs) [33], ES research [34,35], ESs and human well-being [36] and ESs and landscape architecture [37]. Although RLESs have received increasing attention from researchers, they are gradually becoming a research hotspot. However, we have reviewed the literature and found that no researchers have explored sorting out the context and current situation of RLES research through bibliometric research. It is very necessary to fill this research gap.

Therefore, this paper aims to provide a comprehensive knowledge base and systematic overview of RLES research. To help researchers better understand the current state of research as well as directions for future research, we conducted quantitative and qualitative analyses of RLES research through bibliometric methods and visualization tools, assessed the basic features and mapped knowledge areas (e.g., keyword clustering, dual-map overlay and alluvial maps) to identify important topics in RLES research, track the research paths in recent years and make scientific predictions regarding the future direction of the field. We sought answers to the following scientific questions: (1) What are the dynamics of the volume of RLES research publications, which journals have published high-quality research

in this field and which are the most highly cited articles? (2) What are the characteristics of the geographical distribution of collaboration networks, disciplinary distribution and evolution of the knowledge structure of this research? (3) What are the similarities and differences in the focus of research among countries with different economic characteristics, and how has the focus of research changed over time? (4) What are the main current research topics and likely future research trends in the study of RLES research?

## 2. Data Sources and Methods

### 2.1. Data Sources

To ensure that the research data were scientific and comprehensive, the dataset for this paper was obtained from the Science Citation Index Extension (SCI-E) and Social Science Citation Index (SSCI) of the global literature retrieval platform ISI Web of Science Core Collection (WOSCC), which is considered to be the most important and commonly used scientific database in many research areas. The first paper related to RLESs was published in 1990; thus, data extraction began with publications dating back to 1990 to ensure the inclusion of early important research. After several tests, we utilized the advanced search Boolean operator for our study, with the search type constructed as follows: TS = (rural* OR villag* OR countryside*) AND TS = (landsc*) AND TS = (ecosystem* serv* OR ecolo* serv* OR model * OR assessment* OR valu*) (the wildcard "*" means that any word beginning with the preceding letters should be included). The search item was "subject" (covering article title, abstract, author, keyword, keyword Plus and country), only articles and reviews were selected and the search was limited to publications in the English language. A total of 4524 papers were obtained. The publication period was set to 1990–2021, and the data were collected at 9:30 AM on 20 May 2022.

Professor Chaomei Chen has advised that priority should be given to ensuring the completeness of the search and that the completeness of the data was more important than the accuracy of the data [38]. Therefore, given the study's comprehensiveness, we did not carry out any further screening of the literature content (i.e., no further selection based on subject type, subject category or research institutions was made).

### 2.2. Methods

#### 2.2.1. Bibliometric Approach

Bibliometrics is a form of statistical analysis that utilizes visualization tools (e.g., CiteSpace, Mapquetion, Gephi) to perform quantitative and qualitative analyses of academic literature [28,39,40]. This paper uses bibliometrics and quantitative research methods such as mathematical statistics to analyze the number of published papers and journal sources, and it objectively evaluates and reflects the status and development process of a certain field. Through the combination of CiteSpace and Gephi tools, we analyze the influence and cooperation of papers from different countries/regions, as well as the distribution of disciplines and the evolution of knowledge. Based on the co-occurrence cluster analysis of keywords and alluvial map, we identify the evolution of the main research hotspots in this field, and the research focus of countries with different economic levels is obtained. Through literature co-citation cluster analysis, the main research frontiers in this field are ascertained, and emerging trends are identified by burst word detection analysis. Finally, based on the above research, a summary and discussion are presented to predict the future development trends.

#### 2.2.2. Software

The main software used in this paper was CiteSpace, and Gephi, Gis and Mapequation were used for assistance, using visual representation. CiteSpace was developed by Professor Chaomei Chen from the School of Computer and Information Science at Drexel University in the United States based on the Java language [39]. The full name of CiteSpace is Citation Space. It is a citation visualization analysis software that focuses on analyzing the potential knowledge contained in scientific literature and was gradually developed in the context of

scientometrics and data visualization [41]. Since the structure, regularity and distribution of scientific knowledge are presented through visualization, the results obtained via this method are also called "scientific knowledge mapping" [38,42].

*2.3. Research Steps*

This paper presents an overview of RLES research based on bibliometric and knowledge visualization methods, while providing a framework for the literature that can inform future research from the perspectives of macro to micro, intuitive to complex, holistic to local and general to specific. The specific steps carried out are shown in Figure 1.

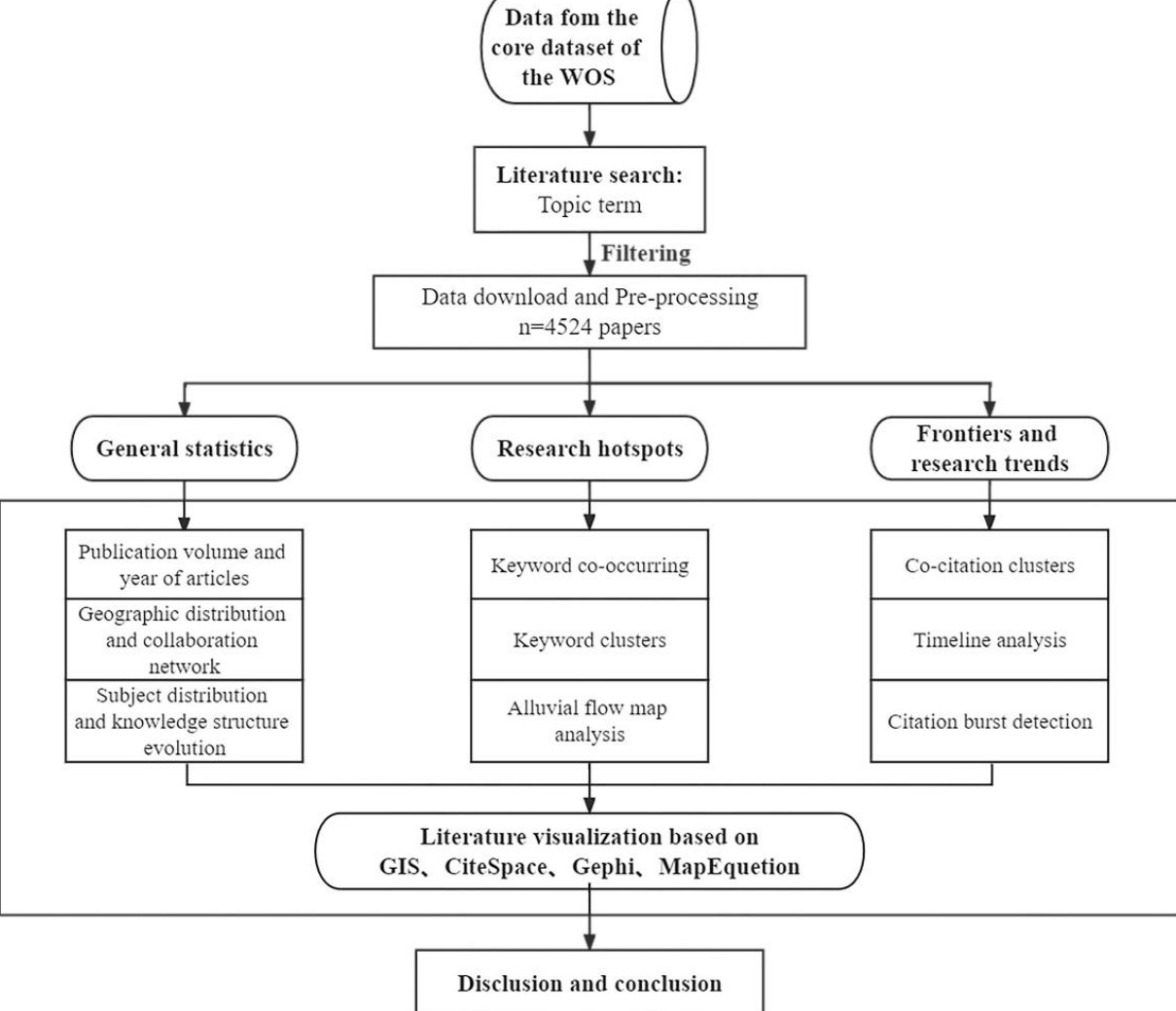

**Figure 1.** Flowchart of the research process.

## 3. Results and Discussion

*3.1. Research Assessment of Rural Landscape Ecosystem Services*

3.1.1. Number of Publications and Journal Sources

Figure 2 shows the total number of WOS publications from 1990 to 2021 (4524), which included 4357 articles and 167 reviews. From 1990 to 2000, the number of publications of RLES-related research was limited, and the field was in its embryonic stage. Early research mainly focused on improving the benefits of ESs through changes in agricultural production structure and land-use policy reforms [43–45]. Between 2001 and 2008, theoretical research on ESs was basically completed, and value assessment became a research hotspot. RLES-related research gradually gained attention and entered its primary

stage; researchers used remote sensing quantitative measurements and the establishment of various models to identify and explore rural landscape elements and the evolution of landscape patterns [46–48]. From 2009 to 2015, the number of publications and citations increased almost steadily and the field was in a stable phase. The number of publications grew the fastest between 2016 and 2021, the selection of quantitative indicators was more abundant, the research scale was more diverse and the research gradually deepened and diversified [49–52], but the number of publications did not exceed 500. The number of citations increased extremely rapidly after 2006, with a total of 20,500 citations in 2021. There were intermittent surges in the number of publications in 2006, 2010 and 2014, and the number of citations exceeded 1000 for the first time. A comparison between RLES and urban landscape ecosystem service (UES) research shows that after 2001, researchers focused more on urban areas, with a wealth of research results. However, less attention was given to ESs in rural areas compared to the boom in urban ecosystem research. This suggests that researchers should focus more on RLESs.

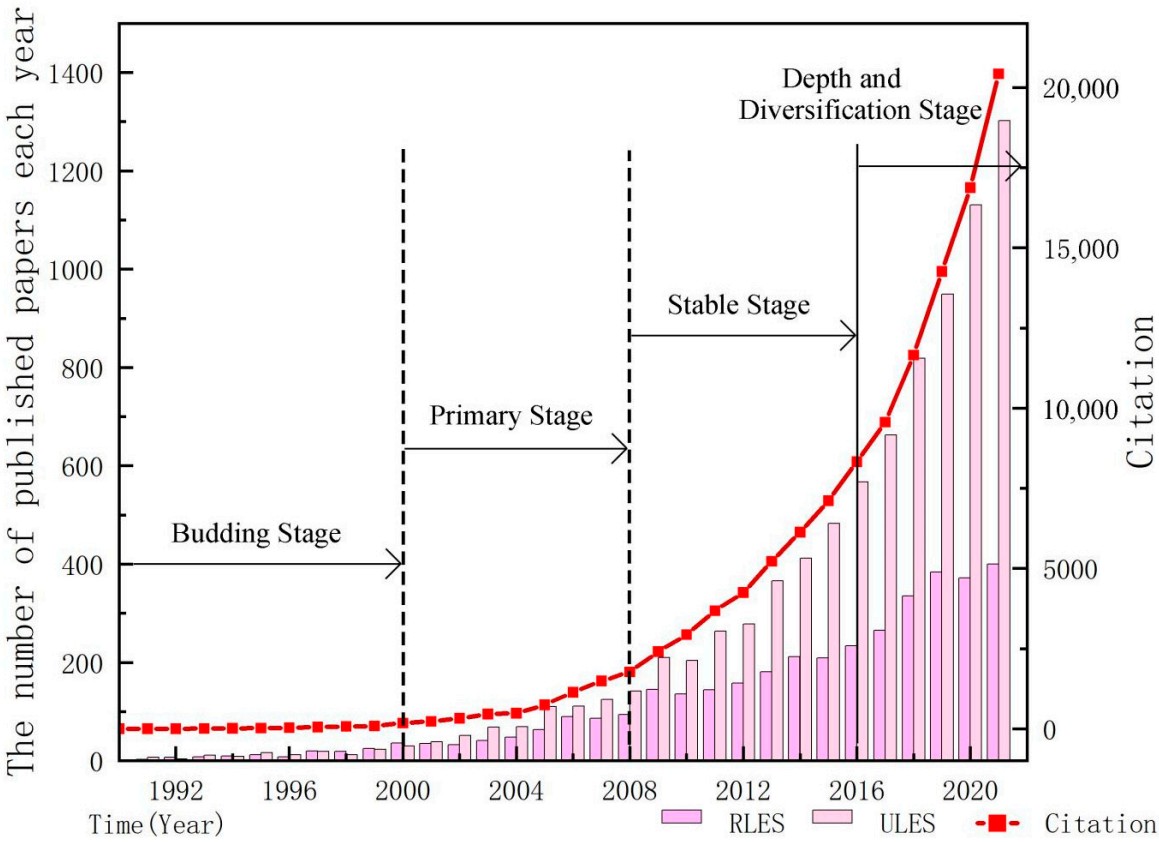

**Figure 2.** Numbers of annual publications and citations from 1990 to 2021. The colored bars indicate different research types.

Citation measurement is a widely used metric for research impact, with higher numbers of citations indicating a more significant research impact. Table 1 lists the top 10 most cited journals in the RLES research literature, showing the number of citations, JCR metrics and the year each journal was first cited. The 4524 articles in this search were published in 209 different journals. The high ranking of 6 of the 10 most cited journals is due to many citations of one or two articles, indicating the high quality and impact of the research in these journals on that research topic.

Table 2 lists the top 10 most frequently co-cited papers from 1990 to 2021, and burst detection was used to confirm these findings (Table 3). The results indicated that the highly cited articles were sufficiently influential and at the forefront of the field. The three articles with the highest citation counts were authored by Bates D (2015) and R Core

Team (2017, 2020). The tools most frequently utilized by researchers for ecosystem service relevance analysis, evaluation and visualization of data were linear mixed models and computational language and environment software for R statistics [53–55]. These three articles were followed by an article by Diaz S detailing the conceptual framework approved by the second plenary meeting of the Intergovernmental Platform on Biodiversity and Ecosystem Services (IPBES); this framework is a highly simplified model of the complex interactions between the natural world and human society most relevant to the IPBES objectives, promoting interoperability among disciplines, stakeholders and knowledge systems [56]. Two articles by Plieninger T looked at drivers of change in European landscapes and assessed, mapped and quantified cultural ecosystem services at a societal level. The first study found that the most prominent direct drivers of landscape change were land abandonment and expansion and that political systems, culture and natural space determined landscape change in different combinations [57]; the second study provided a spatially explicit participatory mapping of the complete range of cultural ecosystem services and several disservices perceived by people living in a cultural landscape in Eastern Germany [58]. Antrop M explained the processes and management of traditional past landscapes and analyzed people's multiple relationships to the perceptible environment and the symbolic meaning that these relationships generate for sustainable planning and management of future landscapes, providing valuable knowledge [59]; it was a landmark article with a central mediator greater than 0.1, and it promoted interdisciplinarity study. Costanza R's review paper was frequently cited in the period between 2018 and 2021. This paper expounds on nearly two decades of ES research, as well as future research directions, suggesting that ESs were at the heart of the fundamental changes needed in economic theory and practice [60]. Van Vliet J systematically reviewed case study evidence on the manifestations and potential drivers of agricultural land-use changes in Europe and found that land-use change trajectories were mainly associated with the transition from rural to urban societies and the transition to post-socialism in Central and Eastern Europe [61].

**Table 1.** Most cited journals in the RLES literature.

| Rank | Source Publication | Impact Factor (JCR 2020) | Number of Citations | Year |
|------|--------------------|--------------------------|---------------------|------|
| 1 | *Landscape Urban Planning* | 6.142 | 1084 | 1993 |
| 2 | *Landscape Ecology* | 3.851 | 911 | 1991 |
| 3 | *Science* | 47.728 | 851 | 1992 |
| 4 | *Biological Conservation* | 5.991 | 786 | 1992 |
| 5 | *Journal of Environmental Management* | 6.789 | 753 | 1992 |
| 6 | *Conservation Biology* | 6.56 | 720 | 1995 |
| 7 | *Agriculture Ecosystems Environment* | 5.567 | 712 | 1993 |
| 8 | *Land Use Policy* | 5.398 | 675 | 1996 |
| 9 | *Nature* | 49.962 | 629 | 1994 |
| 10 | *Proc. Natl. Acad. Sci. USA* | 11.205 | 611 | 2008 |

Number of citations (NC); year indicates the time of the first citation; impact factor (IF): the details were extracted from the journal website.

**Table 2.** The top 10 co-cited references from 1990 to 2021.

| Cited | Co-Cited Reference |
|-------|--------------------|
| 27 | Bates D, 2015 [53], *J Stat Softw*, V67, P1, DOI 10.18637/jss.v067.i01 |
| 25 | R Core Team, 2020 [55], R Lang Env Stat Comp, V0, P0 |
| 24 | R Core Team, 2017 [54], R Lang Env Stat Comp, V0, P0 |
| 18 | Diaz S, 2015 [56], *Curr Opin Env Sust*, V14, P1, DOI 10.1016/j.cosust.2014.11.002 |
| 18 | Plieninger T, 2016 [57], *Land Use Policy*, V57, P204, DOI 10.1016/j.landusepol.2016.04.040 |
| 17 | Antrop M, 2005 [59], *Landscape Urban Planning*, V70, P21, DOI 10.1016/j.landurbplan.2003.10.002 |
| 17 | Plieninger T, 2013 [58], *Land Use Policy*, V33, P118, DOI 10.1016/j.landusepol.2012.12.013 |
| 17 | R Core Team, 2014 [62], R Lang Env Stat Comp, V0, P0 |
| 16 | Costanza R, 2017 [60], *Ecosyst Serv*, V28, P1, DOI 10.1016/j.ecoser.2017.09.008 |
| 16 | van Vliet J, 2015 [61], *Landscape Urban Planning*, V133, P24, DOI 10.1016/j.landurbplan.2014.09.001 |

**Table 3.** Top 10 references with the strongest citation bursts from 1990 to 2021.

| Reference | Title | Strength | Begin | End | 1990–2021 |
|---|---|---|---|---|---|
| Bates D, 2015 [53] | Fitting Linear Mixed-Effects Models Using lme4 | 9.56 | 2017 | 2021 | |
| R Core Team, 2017 [55] | R: A language and environment for statistical computing | 8.49 | 2017 | 2021 | |
| Antrop M, 2005 [59] | Why landscapes of the past are important for the future | 8.85 | 2006 | 2012 | |
| Plieninger T, 2013 [58] | Assessing, mapping, and quantifying cultural ecosystem services at community level | 7.01 | 2014 | 2021 | |
| Diaz S, 2015 [56] | The IPBES Conceptual Framework—connecting nature and people | 6.74 | 2018 | 2021 | |
| Plieninger T, 2016 [57] | The driving forces of landscape change in Europe: A systematic review of the evidence | 6.74 | 2018 | 2010 | |
| R Core Team, 2014 [62] | R: A language and environment for statistical computing | 6.23 | 2015 | 2019 | |
| van Vliet J, 2015 [61] | Manifestations and underlying drivers of agricultural land use change in Europe | 5.99 | 2018 | 2021 | |
| Costanza R, 2017 [60] | Twenty years of ecosystem services: How far have we come and how far do we still need to go? | 5.53 | 2018 | 2021 | |
| R Core Team, 2020 [55] | R: A language and environment for statistical computing | 4.82 | 2020 | 2021 | |

"Strength" represents the intensity of the burst; "begin" represents the starting year of the burst of noun terms; "end" represents the end year of the burst, and the red line represents the duration of the burst.

### 3.1.2. Geographical Distribution and Collaborative Networks

In the context of a knowledge-intensive economy, research is a fundamental variable that determines the development of a country. This paper comprehensively analyzes the geographical distribution and collaborative networks of RLES-related research from 1990 to 2021. The study found 4524 publications representing 142 countries (regions). Figures 3 and 4 show the regional shift in research focus in different periods. Between 1990 and 2005, there were 62 countries/regions, with the central focus regions being the USA, England, Australia, France, the Netherlands and Canada. From 2006–2021, we found that the number of countries studied reached 142, and the study area was almost global. In particular, the number of publications in China has proliferated, with a total of 483 publications. Meanwhile, the study focus area has gradually expanded from developed countries to the world. Table 4 shows the top 10 producing countries (regions) in terms of the number of publications, with the USA consistently being the leading country in this research area with 1115 publications, accounting for 24.66% of the total sum, and making a considerable contribution to the development of RLES research.

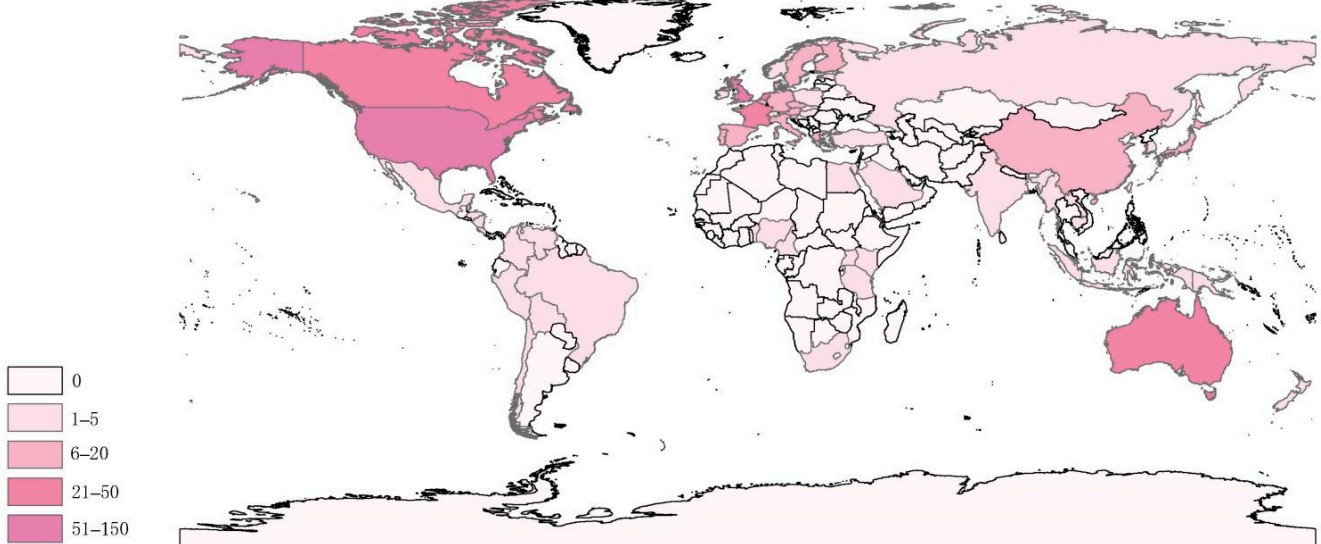

**Figure 3.** Global geographic distributions of publications during the period from 1990 to 2005.

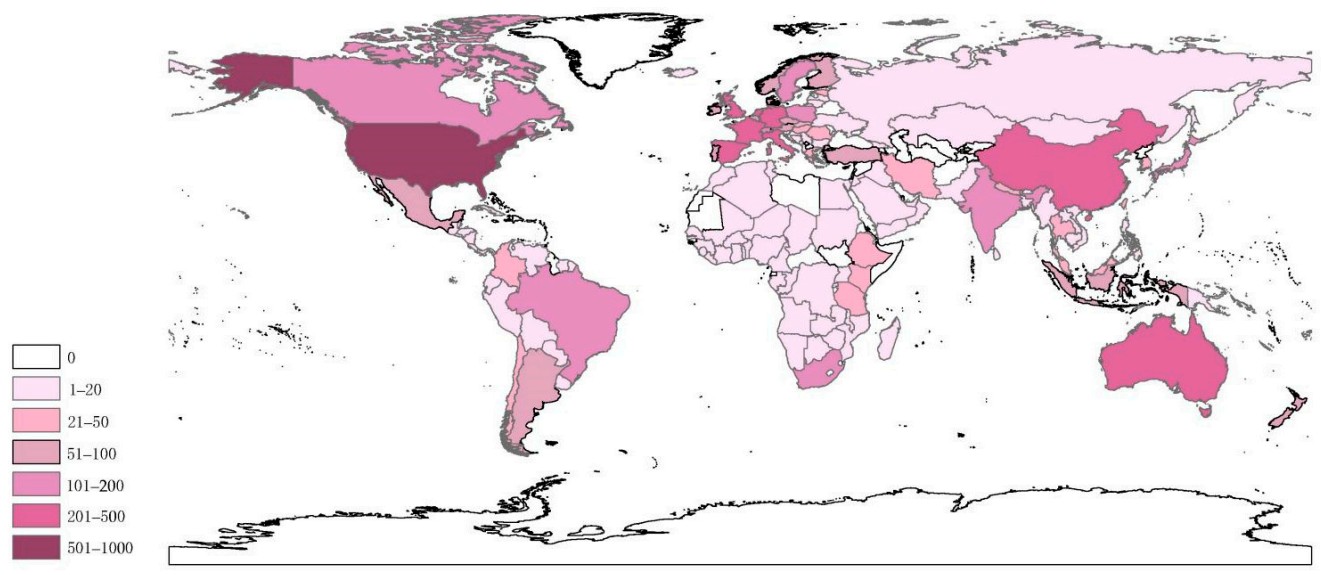

**Figure 4.** Global geographic distributions of publications during the period from 2006 to 2021.

**Table 4.** Top 10 productive countries or regions during the period from 1990 to 2021.

| # | Name | Records | Centrality | # | Name | Records | Centrality |
|---|------|---------|------------|---|------|---------|------------|
| 1 | USA | 1115 | 0.49 | 6 | Australia | 325 | 004 |
| 2 | Peoples R China | 494 | 0.05 | 7 | Spain | 311 | 0.03 |
| 3 | England | 418 | 0.24 | 8 | France | 252 | 0.15 |
| 4 | Italy | 349 | 0.03 | 9 | Netherlands | 252 | 0.11 |
| 5 | Germany | 335 | 0.16 | 10 | Canada | 213 | 0.07 |

Figure 5 shows the six color groups with the USA, England, China, Germany, the Netherlands and Japan as the dominant core, with varying degrees of cooperation. Table 5 shows the top 10 countries in terms of the degree of cooperation, with the USA being the most cooperative, followed by European countries (regions), and China, which had a high number of publications but was not strong in terms of national cooperation. Interestingly, the green and purple areas had the largest cooperation networks and strong and reliable relationships. Almost all members were European countries (regions), which may be due to similar physical and geographical environments. At the same time, European countries (regions) have promulgated some related conventions and policies in the fields of rural revitalization, landscape planning and ecosystem service research, resulting in forming a strong network of research relationships across European countries (regions). However, national (regional) cooperation cannot only be concentrated in a certain region. In the future, every country (region) should join the trend of a global research cooperation network.

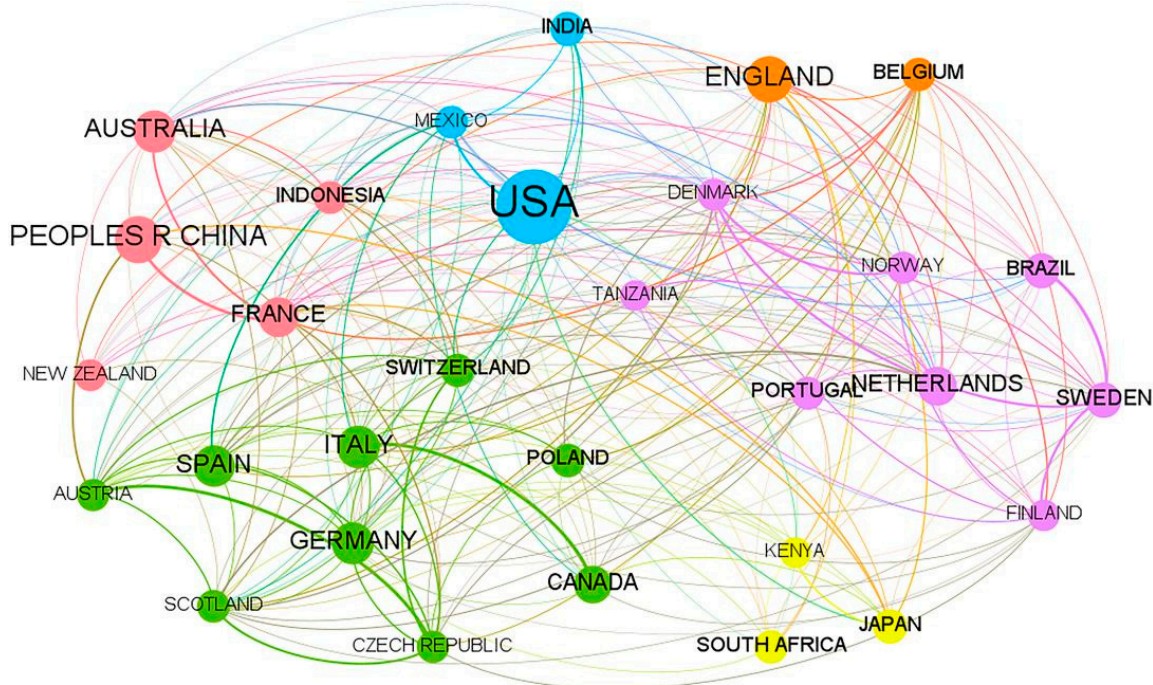

**Figure 5.** Principal cooperation network of productive countries (regions) from 1990 to 2021.

**Table 5.** Top 10 productive countries or regions during the period from 1990 to 2021.

| # | Name | Degree | Year | # | Name | Degree | Year |
|---|------|--------|------|---|------|--------|------|
| 1 | USA | 46 | 1992 | 6 | Sweden | 28 | 2000 |
| 2 | England | 39 | 1995 | 7 | Spain | 25 | 2000 |
| 3 | Germany | 38 | 1999 | 8 | Belgium | 25 | 2000 |
| 4 | Netherlands | 34 | 1994 | 9 | Scotland | 25 | 1998 |
| 5 | France | 33 | 1995 | 10 | Denmark | 25 | 1999 |

### 3.1.3. Distribution of Disciplines and Evolution of Knowledge Structure

The scientific field co-occurrence analysis was conducted by extracting SC fields from the WOS text set and visualizing the results through the "categories" option in CiteSpace. As seen in Figure 6, the analysis showed that the current RLES research literature covers 165 different disciplines, forming a distribution profile with environmental science, ecology, environmental studies, geography and biodiversity conservation at its core and extending to areas such as regional urban planning, forestry and economics. Of these, environmental science and ecology had over 1000 publications related to RLES research. Despite the small number of publications in disciplines such as engineering, anthropology and earth sciences, those publications had a high degree of intermediary centrality and often acted as a bridge for the flow of knowledge among the disciplines in the field.

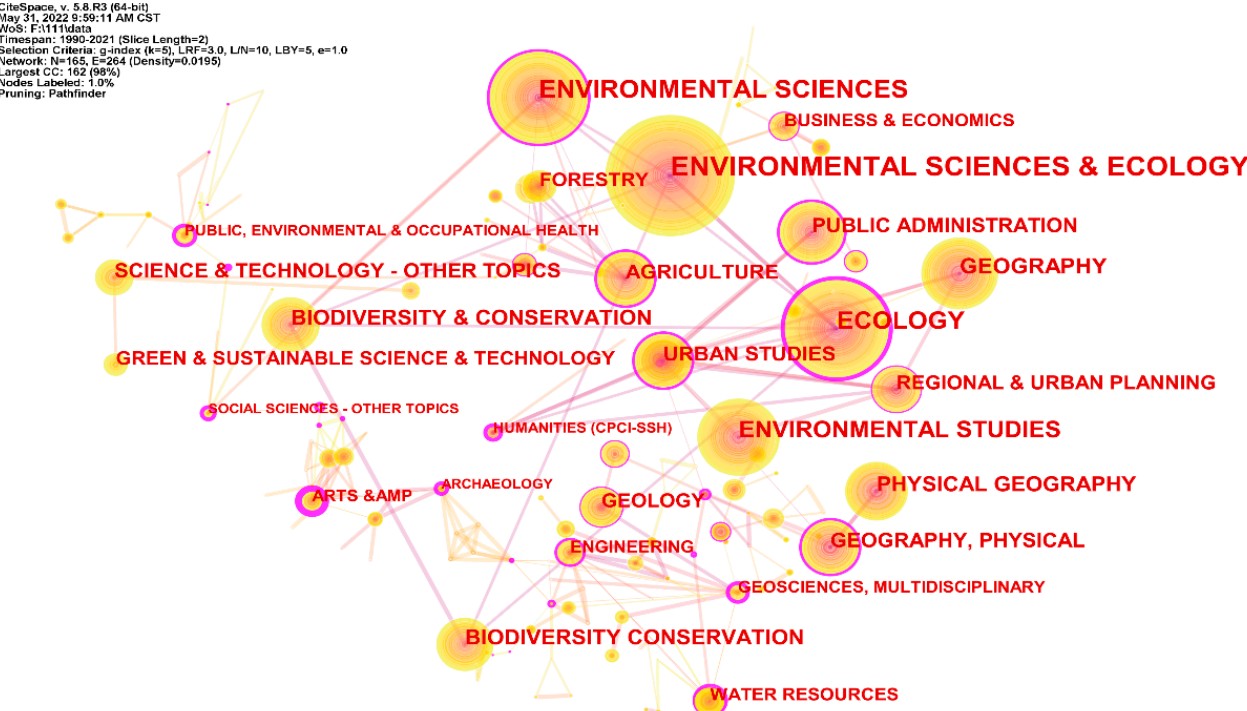

**Figure 6.** Distribution of the main research disciplines from 1990 to 2021.

Using CiteSpace's "JCR Journal Maps" function, dual-map overlays were drawn for the period from 1990 to 2005 and the period from 2006 to 2021. These overlays revealed the evolution of the field's knowledge base by comparing changes in the citation structure of IS journals for each period. As can be seen in Figures 7 and 8, RLES research in the period from 1990 to 2005 was mainly in the disciplines of ecology, geology, oceanography, economics, and political science, and its knowledge base was mainly planted in ecology, zoology, geography, and economic and political science. At this point, the research was not yet interdisciplinary; however, after the period from 2006 to 2021, the research fields of application expanded rapidly (e.g., to animal medicine, psychology, education, health), with one research area corresponding to several knowledge bases and interdisciplinary research.

### 3.1.4. Brief Summary

By analyzing the basic RLES research results, we demonstrate the following: (1) The phased increase in the number of publications is related to the Millennium Ecosystem Assessment (MEA) report released in 2005, the establishment of The Economics of Ecosystems and Biodiversity (TEEB) in 2010, and the establishment of the Intergovernmental Platform on Biodiversity and Ecosystem Services (IPBES). Moreover, others have emphasized that they promote the attention of researchers and policymakers to ESs [60,63,64]. Climate change and accelerated urbanization have led researchers to pay more attention to urban

ecosystem services than rural areas. However, rural ecosystems and cities are an organic whole that is crucial to human well-being and economic development, and research should not neglect the research on rural areas [65]. (2) The number of national articles published and the degree of cooperation are related to countries' natural geographical environment, economic development level, political background and policy promulgation. Future research collaborations should strengthen exchanges between developed and developing countries and promote open science and participatory research [66]. (3) RLESs are still mainly based on environmental science, ecology and economics. There is very little application and research in urban–rural planning, landscape planning and management, and it is still necessary to further strengthen multidisciplinary must be strengthened in the future.

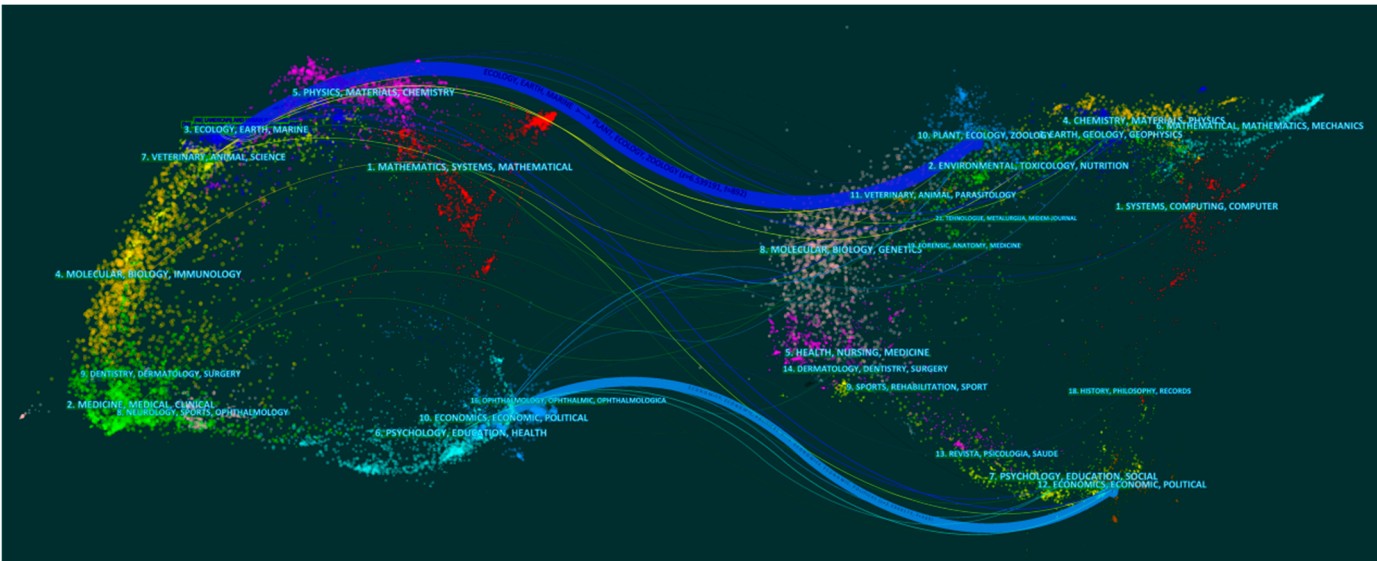

**Figure 7.** A dual-map overlay showing the correlation between disciplines in RLESs during the period from 1990 to 2005.

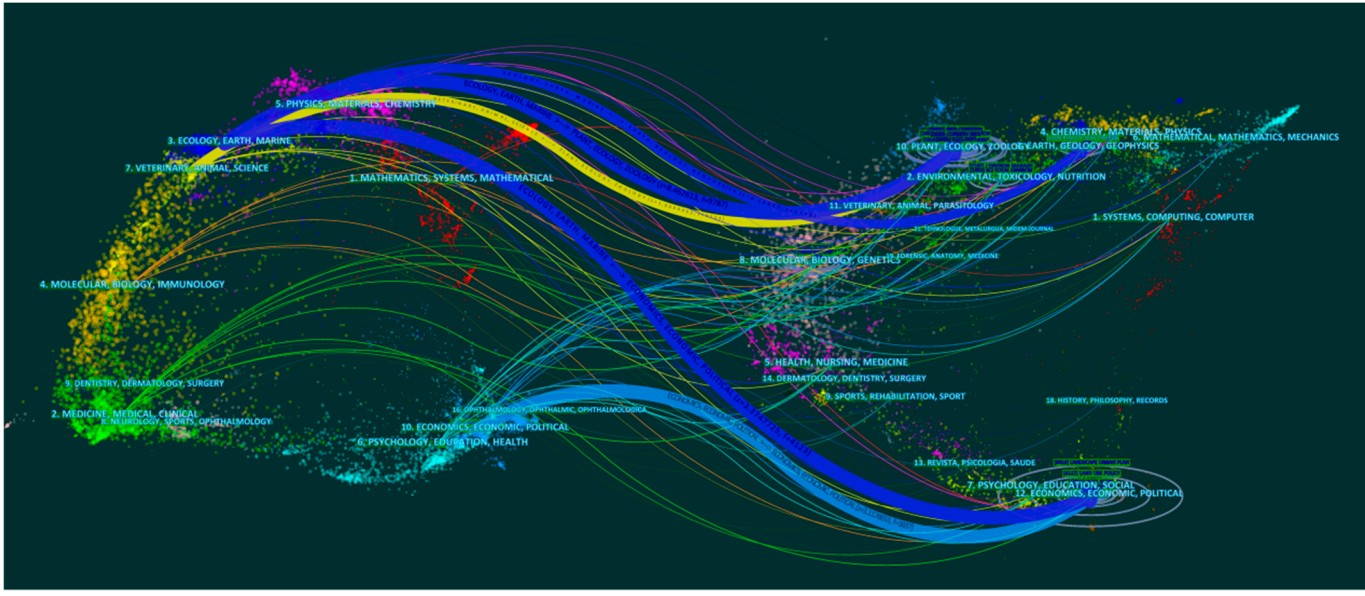

**Figure 8.** Dual-map overlay showing the correlation between RLES disciplines from 2006 to 2021.

*3.2. Hotspots and Frontiers of Research on Rural Landscape Ecosystem Services*

3.2.1. Hotspots for Research on Rural Landscape Ecosystem Services

The keywords in an article often provide a core overview of its topic, so for this paper, keywords were extracted from the literature for co-occurrence analysis to detect popular research hotspots. The countries from which the literature was published in the database were classified into two categories, namely developing and developed countries (according to the 2022 IMF and UN world map of countries by GDP per capita), and keyword clustering analysis was conducted. As seen in Figures 9 and 10, we arrived at 15 and 14 clusters, respectively, and the size of the brown cross indicates the frequency of keywords, such as "landscape", "management", "conservation" and "ecosystem services". The high-frequency keywords reflected the focus of the RLES studies, forming a clustering network structure of closely related keywords, with results indicating a significant and convincing clustering module structure [41]. Details of the clustering labels are shown in Tables A1 and A2.

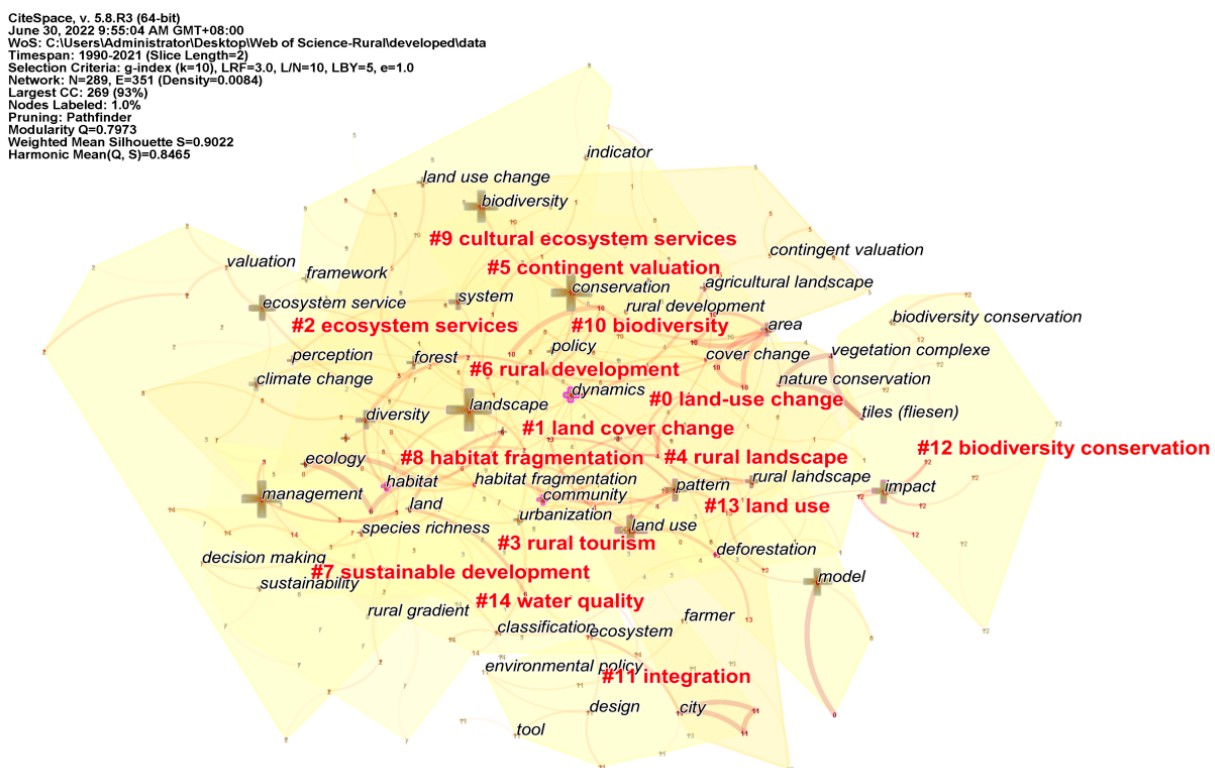

**Figure 9.** Keyword co-occurrence clustering map in developed countries.

According to the keyword co-occurrence cluster analysis, the topics of RLES research in developed countries were mainly focused on "land use cover change", "sustainable rural development", "evaluation", "biodiversity conservation" and "cultural ecosystem services". The main focus of RLES research in developing countries was on "land use change", "biodiversity conservation", "evaluation", "landscape characteristics and indicators" and "spatial heterogeneity". We found many similarities between the RLES research topics in developed and developing countries, including research on "land use change", "biodiversity conservation" and "evaluation". However, due to differences in research backgrounds and other aspects, scholars from developed countries and developing countries had different focuses on research hotspots. From the perspective of research content, in terms of land-use change research, research in developed countries focused on temporal and spatial changes, dynamic mechanisms, policy research and scenario simulation, studying the driving forces of land-use change from the perspective of welfare policy, and carrying out prediction simulation [67–69]. In contrast, scholars in developing countries tended to combine re-

gional characteristics to carry out research on driving mechanism analysis and prediction simulation; at present, integrating agriculture, culture and natural landscape protection to deal with the risk of land-use change has become a vital issue in international research on sustainable rural development [70,71]. From the perspective of research scales, studies in developing countries tended to focus on a single scale, while those in developed countries conducted comparative empirical research on different research subjects from multiple perspectives and dimensions and were relatively rich in microlevel research. At the same time, research in developed countries emphasized the role of technology, knowledge and innovation in RLESs, while in developing countries, research was primarily introductory and tracking-oriented. In addition, RLES research in developing countries, led by China, showed a certain policy orientation and a focus on uncovering problems and phenomena.

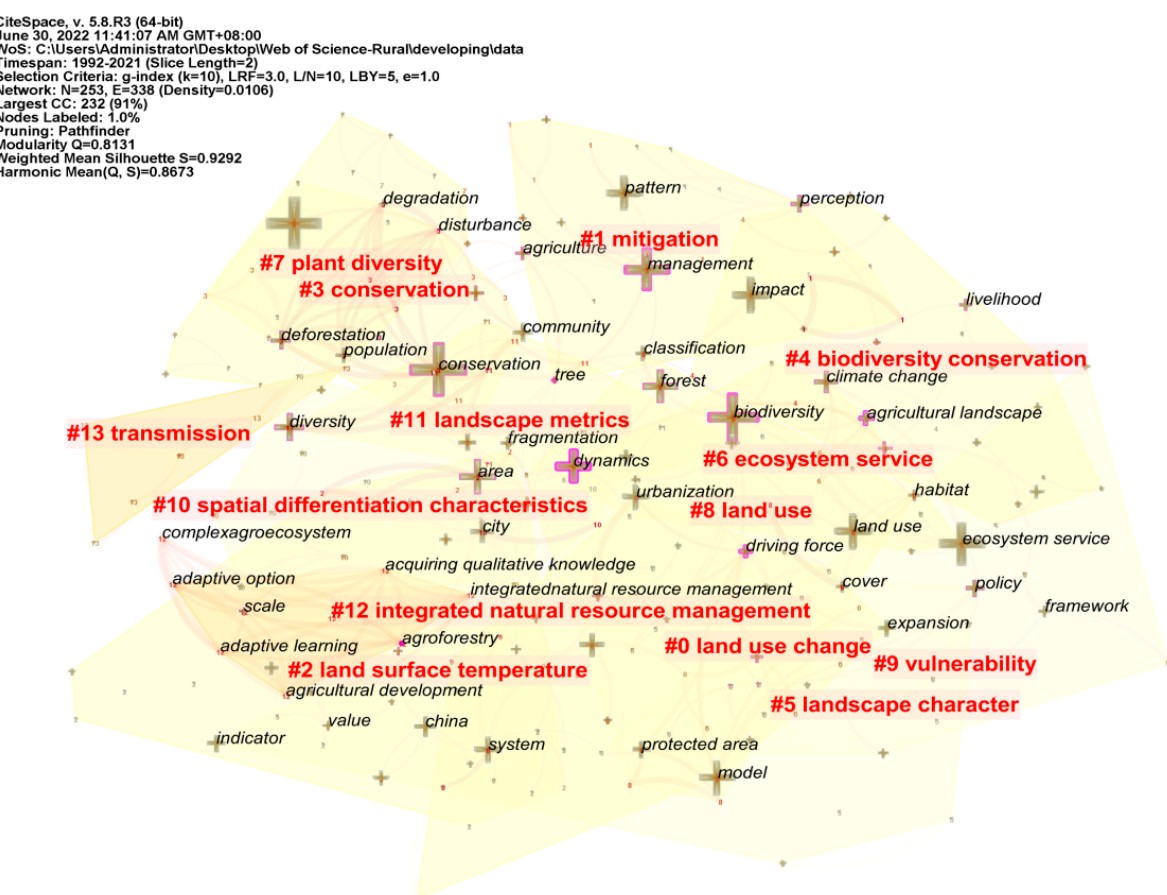

**Figure 10.** Keyword co-occurrence clustering map in developing countries.

By drawing an alluvial map, we observed the evolution of knowledge on RLES research hotspots. The literature was divided into six time periods, keywords from the literature were extracted, and pathfinder networks were generated using CiteSpace and Mapequation [72] (see the work of Rosvall and Bergstrom (2010) for more information on the clustering algorithms and alluvial plots). In this paper, we used page-level values to indicate the importance of keywords, meaning that the higher the page-level value of a keyword was, the higher its frequency was.

The analysis in Figure 11 shows that from 1990 to 1995, the terms "advanced ultrahigh resolution radiometer data", "agriculture", "vegetation" and "land" had relatively high page-level values and the most prolonged alluvial flow period, until 2021. By 1996–2001, the research hotspots had changed to "land use change", "agricultural policy", "rural gradients" and "model", and land-use change research remains a hotspot today. From 2002 to 2007, the number of research themes increased rapidly, with the addition of "management", "rural landscape heritage", "evaluation", "species diversity" and "cultural landscape" to the

previous research hotspots. In the periods from 2008 to 2013 and from 2014 to 2019, there was little shift in the research hotspots, but the focus was on the "structural transformation of agriculture", "spatial patterns", "rural development", "biodiversity" and "human well-being" due to climate change and urbanization. After the period from 2020 to 2021, "cultural ecosystem services", "rural tourism", "landscape preferences" and "policy guidance" emerged as international research priorities. RLES research has had a long history, but it has not focused much on the landscape. The importance of subjective well-being is in the interaction between society and nature, and this has been increasingly recognized in its role [36,73]. It should be noted that the research on rural ecosystem services has a long history since the early days, but has not paid much attention to the landscape. As research topics have become more extensive, scholars have not only expanded assessment methods, models and dynamic properties of RLESs, but also used a cross-mix of research methods across disciplines (e.g., the intersection of ecological, economic and geographic approaches) and, in particular, research on ESs that goes beyond spatial quantification and assessment and is now beginning to help address new questions for policymakers and stakeholders.

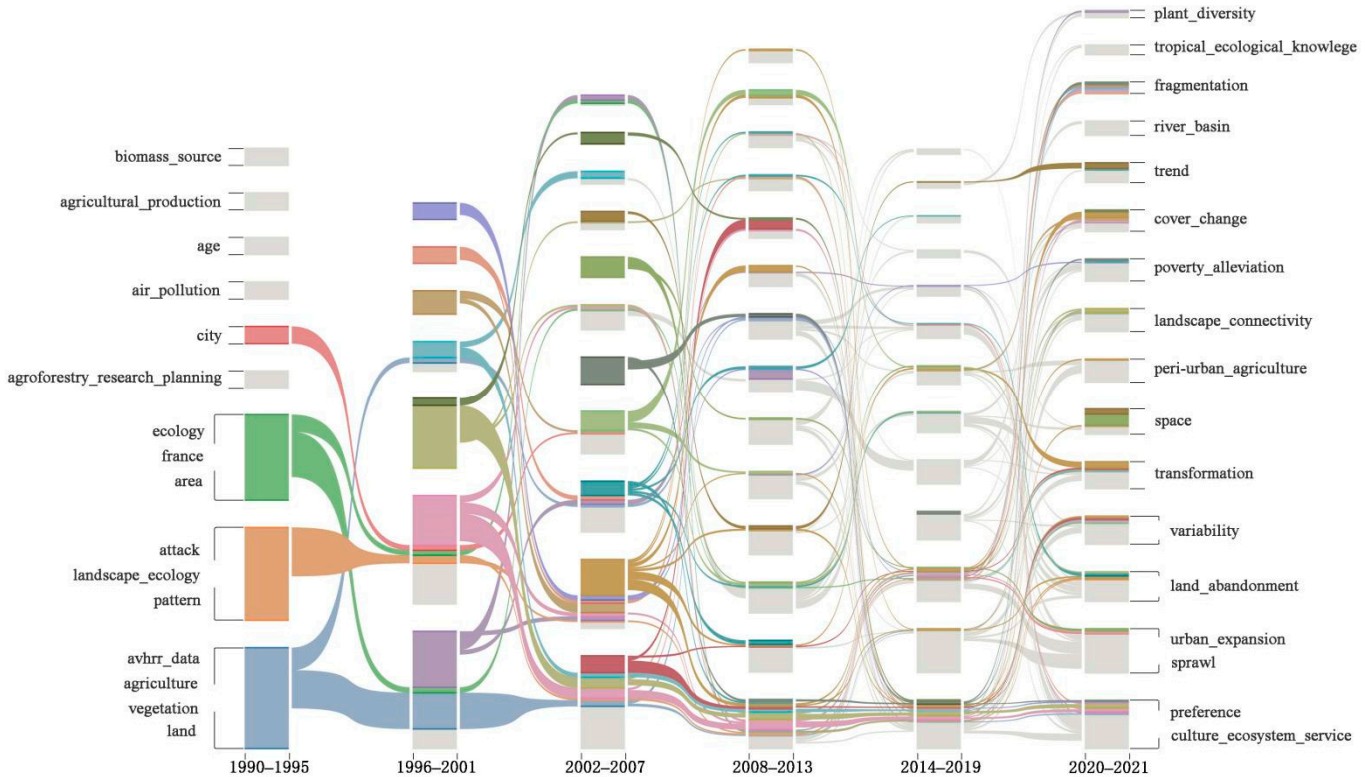

**Figure 11.** Alluvial diagram of research hotspots and progression changes.

### 3.2.2. Frontiers of Research on Rural Landscape Ecosystem Services

A research front is defined as the current state of development in a field, as manifested by the citation clusters, and the co-citation clusters of research frontiers form the field's knowledge base. This paper detected RLES research frontiers through co-citation analysis of the literature, and from Figure 12, we found a total of 16 clusters, with cluster label details shown in Table A3. Combining the above interpretation of research on the foundations of RLES research and the keyword co-occurrence clustering map derived from each branch of research, five categories of research frontiers were summarized. Their main core research contents are sorted out and analyzed as follows:

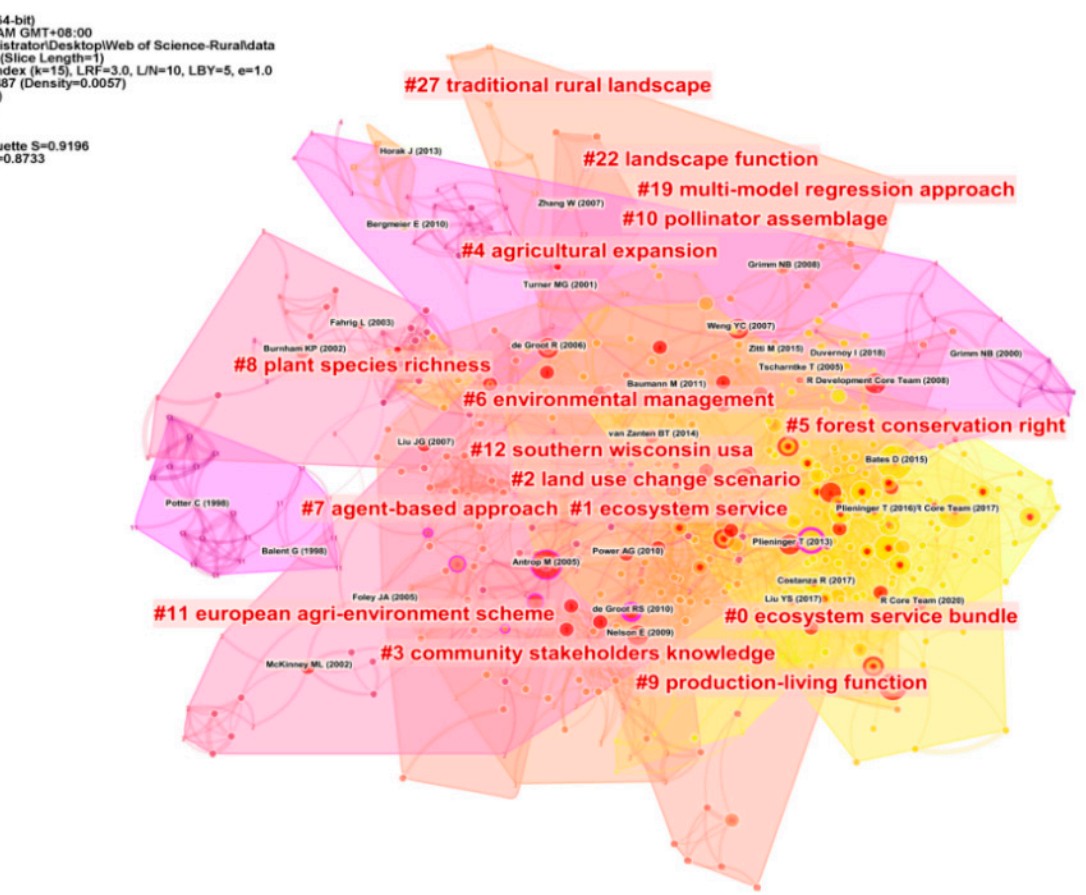

**Figure 12.** Clustering of co-citation references.

1.    Rural agroecological transformation, ecosystem service bundle, land-use planning, drivers

Rural landscapes are currently facing dramatic changes, with the decline of agricultural industries and land-use conflicts and loss of agricultural landscapes becoming increasingly prominent, seriously threatening the sustainability of RLESs. This situation is forcing a transformation of agricultural systems and a structural transformation of rural agroecological systems [74,75]. To systematically reveal the interrelated characteristics of multiple RLESs, some researchers have conducted ES bundle studies on the regional scale, which have mainly entailed clarifying the ecological processes, interrelationships and drivers of various ESs, evaluating and spatially mapping ESs by integrating multiple sources of data, and providing decision makers with optimal ecosystem pattern combination studies [10,76,77]. On the other hand, the importance of land use in influencing the range of ESs provided by rural landscapes has been increasingly recognized by quantifying the economic impacts of land-use change on ESs and conducting land-use planning, which helps to advise policymakers and improve rural ecological sustainability [69,78,79].

2.    Ecosystem service supply and demand, evaluation, spatial quantification and valuation, multiscale models

To reduce and alleviate the trade-off and imbalance between the supply of and demand for ESs, researchers have used techniques such as constructing indices and assessments, scenario development and simulation and spatial mapping and analysis and have captured the mechanisms, expressions and fundamental characteristics of supply and demand relationships between various types of ESs at various spatial and temporal scales. This has been done to lessen and mitigate the trade-offs and imbalances between the supply and demand of ESs [68,80–82]. There has been a strong trend towards research into the perception and valuation of RLESs, with comprehensive valuation providing key information to support decision-making in the management of rural ecosystem services.

It is applied in the evaluation of ESs such as agro-ecological transformation, forestry protection management and guidance of agricultural production but less in rural landscape planning and management [83–88]. Furthermore, the most dominant approach to value evaluation has been monetary valuation, and its tools are various, but it will be important to pay attention to real needs beyond currency valuation [60,89,90].

3.  Cultural ecosystem services, stakeholders, policy development, regional scale

Cultural services are one of the four categories that make up the classification of common ESs, linking the bridges that connect social systems to natural systems. Their mediation potential is low compared to supporting and regulating services [3]. However, when an area's provisioning and regulating services are degraded, they are replaced by socioeconomic instruments, but their cultural values are irreplaceable. Research on cultural ecosystem services has mainly focused on six main areas, monetization evaluation, management and application, indicator systems, value mapping, recreational function and aesthetic functions of cultural service value. Research shows that landscape planning involves stakeholders at all scales, including policy makers and implementers, land users and those receiving services, helping to assist in ecosystem-related decision-making, which improves the biodiversity, ecological function, landscape function and ecosystem service capacity of rural areas [83,91,92].

4.  Biodiversity and ecosystem services synergies, trade-offs and management, climate change, human well-being

Biodiversity and ESs are the material basis for human survival and sustainable socioeconomic development and are critical to human well-being. Addressing the loss of biodiversity and the degradation of ESs has become another global environmental hotspot amidst climate change [93,94]. As an area of high biodiversity, the countryside must rationally allocate ecological assets, conserve biodiversity and provide adequate ESs within it, becoming a challenge that must be faced in the sustainable development of rural landscapes. Research has shown that trade-offs and synergies between biodiversity and ecosystem services can better help managers make more beneficial decisions, protect rural ecosystems and reduce biodiversity by formulating rational planning and adaptation strategies for the adverse impacts of crises. To determine the spatial pattern of ESs and biodiversity, rural ecosystem management can be carried out effectively and sustainably [93].

5.  Urban–rural synergistic development, biodiversity management, rural landscape conservation, landscape structure

Amidst the impacts of urbanization, the degradation of ESs and changes in landscape structure have made rural sustainable development and biodiversity conservation critical. To address this issue, several studies have found that landscape connectivity between urban and rural areas can provide favorable habitats for certain species and that innovative approaches to coupling landscape structure and ecosystem services (LS-ES) along urban–rural gradients provide potent tools for urban–rural ecosystem planning and synergistic development [95,96]. Many scholars are gradually shifting their focus to strategies for the coordinated development of urbanization, rural landscapes and biodiversity. The dynamic changes in urbanization have brought new patterns to rural landscapes that facilitate rural biodiversity conservation and ecosystem service functions by enhancing landscape connectivity between urban and rural areas and providing favorable habitats for certain species, which are beneficial for improving rural biodiversity conservation and ESs [97–99].

### 3.2.3. Brief Summary

By analyzing the research hotspots and frontiers of RLES research, we demonstrate that "land use change", "biodiversity conservation" and "value assessment" are both the research hotspots and frontiers of RLESs, and the research will continue in the future. At the same time, previous studies have shown that these three aspects are also hotspots in the research fields of UESs [30], human health and well-being [36] and AESs [100], which shows

that research in these areas is critical. We also demonstrate that, although "landscape" is more frequent than "ecosystem services", the degree of association between the two is very low. This phenomenon is attributed to the expression of keywords, and researchers seldom apply ES research to landscape planning in rural areas and consider landscape factors less. By drawing alluvial maps to analyze the evolution of research hotspots, we found that the field is gradually shifting from the traditional content of taxonomic and theoretical studies to the analysis of multiple spatial and temporal scales, analysis of different service trade-offs and synergies and analysis of dynamic evolutionary processes. It is worth noting that current research focuses more on the perception of subjective well-being, including aspects such as "cultural ecosystem services", "human well-being" and "landscape preferences" [73,84,101]. We know that the ultimate purpose of ES research is to translate science into practical policy decisions; although there are many research findings that have been applied to decision-making and are sufficiently specific and instructive at the national level [85,102], they are still too broad to be applicable to ecological management in local and cross-administrative regions, and research findings are not effectively translated into decision support tools to address policy issues [82,103]. Therefore, we need to truly understand whether ES research projects affect decision-makers, propose new decision support transformation tools and involve non-material benefits of cultural ecosystem services in decision-making, which can improve the social recognition and legitimacy of management decisions, as well as promoting the high integration of ES-related research with management decision-making and policy design and the improvement of human well-being.

### 3.3. Trends in Research on Rural Landscape Ecosystem Services

3.3.1. Burst Word Detection

Burst word detection is a computational technical tool to identify sudden events or important information, and more vigorous bursts indicate higher interest in that relevant research topic which is likely to be the focus of future research. In this study, we extracted noun terms from the top 100 articles in RLES research over the last decade, visualized and analyzed the terms with the strongest citation bursts and used them to predict future research trends.

Table 6 shows the top 20 emergent terms with the strongest citation bursts. We found that research topics related to human well-being and land-use change were popular in RLES research from 2012 to 2021, and multifunctional landscape studies became a hot topic in the last five years. In terms of timing and content, previous research focused on structural changes in rural landscapes, identification of landscape elements and vegetation type studies. The issue of land abandonment and declining ESs due to climate change prompted the structural transformation of agriculture, and the number of studies exploded over the five years ending in 2017. Since 2018, amidst the issues with the rural population and urbanization, the need for diversity in rural landscapes has increased, and multifunctional landscapes have become a significant research area. Current researchers are gradually focusing on urban–rural synergies and the quality of ESs, improving assessment methods and focusing on research related to human well-being and health.

**Table 6.** Top 10 productive countries or regions from 1990 to 2021.

| Title | Strength | Begin | End | 2012–2021 |
|---|---|---|---|---|
| human health | 7.9 | 2019 | 2021 | |
| spatial scales | 5.88 | 2015 | 2018 | |
| rural population | 5.7 | 2019 | 2021 | |
| urban growth | 5.28 | 2015 | 2018 | |
| functional diversity | 5.26 | 2019 | 2021 | |
| ecological process | 5.15 | 2014 | 2016 | |
| linear mixed model | 5.14 | 2014 | 2017 | |
| positive effect | 4.96 | 2017 | 2018 | |

**Table 6.** *Cont.*

| Title | Strength | Begin | End | 2012–2021 |
|---|---|---|---|---|
| landscape element | 4.59 | 2012 | 2014 | |
| vegetation type | 4.59 | 2012 | 2014 | |
| landscape variable | 4.54 | 2012 | 2014 | |
| land abandonment | 4.44 | 2013 | 2017 | |
| major challenge | 4.38 | 2019 | 2021 | |
| urban planners | 4.38 | 2019 | 2021 | |
| local level | 4.31 | 2015 | 2017 | |
| land uses | 4.29 | 2013 | 2014 | |
| agricultural lands | 4.29 | 2018 | 2019 | |
| tree species | 4.28 | 2018 | 2019 | |
| regional scales | 4.28 | 2015 | 2017 | |
| landscape variable | 7.9 | 2019 | 2021 | |

"Strength" represents the intensity of the burst; "begin" represents the starting year of the burst of noun terms; "end" represents the end year of the burst, and the red line represents the duration of the burst.

### 3.3.2. Future Research Trends

Based on previous fundamental research and analysis of research hotspots and frontiers, we predict that RLES research area will focus on four areas in the future. The first is relationships and collaboration among and management of biodiversity and ESs. We need to investigate the characteristics, drivers and trends of biodiversity and ESs; establish service value assessment models; and strengthen the links with policy and management in the future [104,105]. Through comprehensive disciplines, such as ecology, economics, sociology and management, we must use these principles to propose methods to improve ESs and balance patterns of access to and distribution of RLES research for various ecosystem benefits to promote biodiversity conservation and enhance biodiversity human health and well-being.

Second, we must study the landscape value of rural cultural ecosystem services. As social and economic development and the level of human needs increase, cultural ecosystem services will play an increasingly significant role in enhancing human well-being. There are still cognitive differences and methodological shortcomings in the study of cultural ecosystem services, and their systematic study is lacking. At the same time, MEA (2005) argued that cultural services and values were not sufficiently recognized in landscape planning and management. Assessment methods and spatial quantification and valuation still face difficulties, as inconsistencies in assessment scales hinder the comparability of cultural ecosystem studies conducted in different disciplinary contexts [106–108]. Therefore, future research should give more consideration to the value and connotation of landscape in cultural ecosystem services, focusing on different types of cultural ecosystem services [101,109] and paying attention to in-depth communication among multidimensional stakeholders. Through scale conversion, we can comprehensively understand the linkages between different scales, including interactions with other ecosystem services in rural landscapes at different scales and interactions of different ESs at the same scale.

Third, research should be conducted on land-use change and the value of ESs. Researchers should pay more attention to the impact of the former on the latter and other aspects of the relationship between them, strengthen research on dynamic assessment methods and spatiotemporal dynamic assessment models of ESs, enhance research on the ecological mechanisms of the impact of land-use change on ESs and focus on the bundle application of ES assessment results amidst land-use change [110,111]. The ultimate aim of ES valuation research is to translate science into practical policy decisions, and we found that some services have been undervalued in the decision-making process; hence, this will be a focus of future research [112–114].

Finally, with the development of information and the era of big data, new technologies and tools continue to be applied across disciplines, and innovation in research methods is crucial for RLES research. For example, the open statistical software R is increasingly

used in academia [54] and can facilitate interdisciplinary research. In addition, economic or statistical models can improve data analysis [53,115,116]. We need to integrate research methods from various disciplines and develop new methods for quantifying ecosystem multifunctionality and ecosystem multiservice based on an understanding of the strengths and weaknesses of existing methods and their mathematical rationale. In the future, we need to innovate spatial assessment technologies that directly match practical needs; develop indicators and assessment technologies that are directly related to people's well-being; and strengthen the intersection of 3S (GIS, GPS, RS), TM image interpretation, statistical analysis and spatial analysis. By utilizing and continuously expanding the research ideas on the dynamic valuation of ESs, it is possible to study the dynamics of change in rural landscape ecosystems and the dynamic valuation of their value.

### 3.3.3. Brief Summary

Through burst word detection analysis, we predict that future research will mainly focus on four aspects: strengthening the relationships between biodiversity, land-use change and ESs; value assessment; collaboration and management research; and multifunctional landscape research and cultural ecosystem services in rural landscapes. Likewise, other researchers make the same predictions [83,110,117–119]. However, it is necessary to pay attention to the research on landscape value. Some researchers have indicated that more consideration should be given to the value and connotation of landscape in cultural ecosystem services and should focus on different types of cultural ecosystem services [101]. We need to scientifically and comprehensively perceive and evaluate the characteristics, supply and demand, patterns and values of RLESs from multiple scales, angles and methods and strengthen innovation in ecosystem service assessment indicators so that they are directly related to people's values and needs, rather than purely ecological indicators, biophysical indicators or common socioeconomic indicators. Better linkages are needed between theoretical research, practical application and decision management. Collaboration and analysis across multiple disciplines, professions and regions should continue into the future. With the cross-integration of disciplines and the trend of cross-administrative, cross-regional and cross-country research, researchers should focus on comprehensive applications.

### 3.4. Limitation of This Study

The study has some limitations. First, we only collected SCI-E and SSCI publications from the WOS Core Collection database and did not use other databases to obtain a larger sample of studies. Future studies should also include other databases, such as Scopus and Google Scholar, to verify the findings of this study. We selected only articles published in English for analysis, and some papers written in other languages than English were excluded. Second, only the LLR algorithm was used for cluster resolution in the scientific knowledge mapping analysis. However, future studies should combine the three association-analysis algorithms to improve accuracy. Again, this study focused on analyzing and interpreting the focused clusters. We did not conduct a further analysis of the nonfocused clusters due to the textual limitations and the insignificant research pathway of some clusters. The research frontier and future development trends were identified based on a software analysis. The results were subjective, and further reading and collation of the literature are needed.

## 4. Conclusions

This paper uses a bibliometric approach to evaluate the basic characteristics, hotspots, frontiers and future trends of RLES research over the last 30 years (1990–2021) through qualitative and quantitative analysis. It is expected to provide a theoretical basis for future RLES-related research; help managers and policymakers in planning, decision-making and protection; and improve the sustainable development of rural landscapes and human well-being. We reach the following conclusions:

(1)　Research in this field is divided into four stages: from 1990 to 2000, the embryonic stage; from 2001 to 2008, the primary stage; from 2009 to 2015, the stable stage; and from 2016 to 2021, the number of publications grew the fastest, and the research quickly deepened and diversified. At the same time, the study found that scholars paid more attention to urban areas, with less attention given to rural ecosystem services. This phenomenon suggests that researchers should pay more attention to RLES research. Studies have shown the increasing availability and accuracy of algorithms and tools, which has prompted quantitative evaluation research related to RLESs.

(2)　The 4524 publications in the database represent 144 countries/regions, of which the United States is the most prolific country. The main research countries also include China, England, Italy and Germany. The United States and European countries (regions) have the highest degree of cooperation. They form a strong and stable network of research collaborations. From the analysis of the distribution of disciplines and the evolution of knowledge structure, it can be found that RLES research is interdisciplinary research, but that the main disciplines are still environmental science, ecology, economics, political science, etc., with very few applications in urban–rural planning, landscape planning and management.

(3)　The RLES research hotspots in developed countries mainly focus on five aspects: "land use cover change", "rural sustainable development", "value assessment", "biodiversity conservation" and "cultural ecosystem services". Developing countries mainly focus on "land use change", "biodiversity protection", "value assessment", "landscape characteristics and indicators" and "spatial heterogeneity characteristics". Due to differences in research background and other aspects, developed countries are more abundant in research content and scale than developing countries. By making alluvial maps, we found that research topics are broadening, moving from earlier studies such as "advanced ultrahigh-resolution radiometer data", "agriculture", "vegetation" and "land" to "cultural ecosystem services", "rural tourism", "landscape preference" and "policy guidance".

(4)　Through literature co-citation analysis, we have summarized five research fronts. These include rural agroecological transformation, ecosystem service integration, land-use planning and drivers; using multiscale models to study rural ecosystem service supply and demand, value assessment, spatial quantification and valuation; monetizing cultural ecosystem services in rural landscape assessment, indicator system establishment and landscape value research, involving stakeholders of different scales to assist in decision-making related to ESs; research on the synergistic relationship between biodiversity and ESs, ESs and human well-being; and paying attention to the coordinated development of urban and rural areas and protecting the rural landscape and landscape structure.

(5)　Through detecting and analyzing burst words, we found that researchers are now paying more attention to multifunctional landscape research, improving the coordinated development of urban–rural areas and the quality of ESs, improving value assessment methods and paying attention to related research, such as in human well-being and health. Finally, based on previous basic research and analysis of research hotspots and frontiers, we predict that future research will mainly focus on four main aspects: the relationship between biodiversity and ESs, collaboration and management research; research on the landscape value of rural cultural ecosystem services; land-use change and ecosystem service value research; innovative research methods for RLES research.

**Author Contributions:** Conceptualization, Y.W., B.X. and Y.Z.; data curation, Y.W. and Y.Z.; formal analysis, Y.W., G.Y. and Y.Z.; funding acquisition, B.X.; investigation, Y.W. and X.C.; methodology, G.Y. and B.X.; software, Y.Z., Y.W. and J.W.; validation, B.X. and Y.W.; visualization, Y.Z. and B.X.; writing—original draft, Y.Z. and Y.W.; writing—review and editing, Y.Z., B.X., G.Y., J.W. and X.C. All authors have read and agreed to the published version of the manuscript.

**Funding:** This research was funded by the Key Research and Development Program of the Zhejiang Provincial Department of Science and Technology, grant number 2019C02023.

**Institutional Review Board Statement:** Not applicable.

**Informed Consent Statement:** Not applicable.

**Data Availability Statement:** Not applicable.

**Acknowledgments:** This research was financially supported by the Key Research and Development Program of the Zhejiang Provincial Department of Science and Technology, grant number 2019C02023. The authors would like to thank the anonymous reviewers for their helpful and constructive comments.

**Conflicts of Interest:** The authors declare no conflict of interest.

## Appendix A

**Table A1.** Analysis of research topics in developed countries based on keywords.

| Cluster ID | Silhouette | Hotspots (LLR) | Main Keywords | Numbers of Keywords |
|---|---|---|---|---|
| #0 | 0.934 | land-use change | model; dynamics; driving force; climate; spatial pattern; politics; landscape dynamics; land abandonment; vulnerability; cover change | 24 |
| #1 | 0.917 | land cover change | policy; resilience; cultural landscape; land cover change; trend; Europe; future; countryside; consequence; simulation | 33 |
| #2 | 0.9 | ecosystem services | landscape; ecosystem services; diversity; habitat; framework; valuation; agricultural intensification; behavior; health; driver | 22 |
| #3 | 0.728 | rural tourism | management; growth; fragmentation; rural area; knowledge; landscape pattern; habitat use; richness; exposure; heritage | 21 |
| #4 | 0.914 | rural landscape | community; rural landscape; population; social-ecological system; nature conservation; dispersal; participation; economics; farmland; connectivity | 21 |
| #5 | 0.92 | contingent valuation | forest; agricultural landscape; preference; vegetation; contingent valuation; land use change; scale; benefit; agri-environment program; demand | 19 |
| #6 | 0.899 | rural development | ecology; land; rural development; challenge; rural gentrification; geography; restoration; risk; governance; perspective | 19 |
| #7 | 0.823 | sustainable development | climate change; sustainable; decision making; rural livelihood; poverty alleviation; protected area; conflict; demography; carbon sequestration; opportunity | 18 |
| #8 | 0.847 | habitat fragmentation | species richness; habitat fragmentation; landscape context; rural hemiboreal landscape; geographic information system; movement; rural gradient; degradation; extinction; shifting cultivation | 17 |
| #9 | 1 | cultural ecosystem services | system; indicator; perception; trade-off; attitude; value; strategy; complexity; landscape management; place attachment | 16 |
| #10 | 0.941 | biodiversity | conservation; biodiversity; area; bird; transition; anopheles albimanus; agricultural land; home range; America; logistic regression | 15 |
| #11 | 0.936 | integration | farmer; ecology; design; city; tool; environment policy; green space; emission; agroindustrial building; landscape ideal | 14 |

**Table A1.** *Cont.*

| Cluster ID | Silhouette | Hotspots (LLR) | Main Keywords | Numbers of Keywords |
|---|---|---|---|---|
| #12 | 0.939 | biodiversity conservation | impact; environment; cover; biodiversity conservation; nation park; expansion; regression; determinant; disturbance | 14 |
| #13 | 0.955 | land use | land use; pattern; deforestation; landscape change; livelihood; landscape ecology; index; attack; plantation aerial photography interpretation | 13 |
| #14 | 0.949 | water quality | urbanization; classification; remote sensing; water quality; tree; biomass; catchment; urban heat island; runoff; nitrogen | 13 |

**Table A2.** Analysis of research topics in developing countries based on keywords.

| Cluster ID | Silhouette | Hotspots (LLR) | Main Keywords | Numbers of Keywords |
|---|---|---|---|---|
| #0 | 0.94 | Land-use change | model; system; urbanization; driving force; expansion; spatial pattern; landscape change; cover change; rural settlement; sustainable development | 30 |
| #1 | 0.944 | mitigation | management; impact; pattern; perception; agriculture; rural development; knowledge; performance; density; preference | 26 |
| #2 | 0.962 | land surface temperature | area; remote sensing; region; China; determinant; land cover; landscape pattern; grown; time series; government | 24 |
| #3 | 0.993 | conservation | population; conservation; community; vegetation; ecology; deforestation; emission; productivity; degradation; analytical framework | 21 |
| #4 | 0.827 | biodiversity conservation | environment; trade-off; biodiversity conservation; strategy; challenge; livelihood; multifunction landscape; payment; environment service; design | 17 |
| #5 | 0.859 | landscape character | indicator; policy; climate; quality; value; perspective; transformation; spatial distribution; settlement; support | 16 |
| #6 | 1 | ecosystem service | dynamics; climate change; agricultural landscape; decision making; benefit; demand; biological invasion; cultural ecosystem service; functional diversity; carbon | 15 |
| #7 | 0.967 | plant diversity | rural area; green space; species richness; consumption; bushmeat; habitat fragmentation; plant diversity; human-carnivore conflict; habitat suitability; extinction | 14 |
| #8 | 0.953 | land use | land use; biodiversity; forest; classification; protected area; habitat; cover; valuation; stakeholder; environment impact | 14 |
| #9 | 0.834 | vulnerability | framework; vulnerability; poverty alleviation; service; adaptation; rural livelihood; restoration; assemblage; resilience; complexity | 14 |
| #10 | 0.863 | spatial differentiation characteristics | city; sustainability; evolution; urban; farmer; Africa conflict; driving mechanism; attitude; ecosystem service | 13 |
| #11 | 0.896 | landscape metrics | diversity; fragmentation; land; landscape metrics; connectivity; cultural landscape; prediction; forest policy; Atlantic forest; abundance | 13 |

**Table A2.** *Cont.*

| Cluster ID | Silhouette | Hotspots (LLR) | Main Keywords | Numbers of Keywords |
|---|---|---|---|---|
| #12 | 0.993 | integrated natural resource management | scale; agroforestry; agricultural development; adaptive option; integrated natural resource management; acquiring qualitative knowledge; adaptive learning; complex agroecosystem; actor-oriented approach | 9 |
| #13 | 0.959 | transmission | prevalence; infection; transmission; disease; risk factor; epidemiology | 6 |

**Table A3.** Top 16 clusters by size based on co-citation references.

| Cluster ID | Size | Silhouette | Top Term (LLR) | Mean (Year) |
|---|---|---|---|---|
| #0 | 110 | 0.809 | ecosystem service bundle; land-use planning; ecosystem service value; agricultural landscape; drivers consequence | 2015 |
| #1 | 80 | 0.857 | ecosystem service; ecosystem service provision; mapping tool; supporting management; Mediterranean agroecosystem | 2011 |
| #2 | 58 | 0.961 | Land-use change scenario; delta economic area; socioeconomic driving force; Yangtze river; environmental processes | 2003 |
| #3 | 44 | 0.958 | community stakeholders knowledge; mapping indicator; landscape assessment; multiscale modelling approach; analyzing landscape service dynamics | 2008 |
| #4 | 38 | 0.986 | agricultural expansion; social access; Spatiotemporal pattern; northeast Thailand; historical patch-level analysis | 2000 |
| #5 | 37 | 0.913 | forest conservation rights; ecosystem service; landscape diversity; conceptual link; diet diversity | 2016 |
| #6 | 36 | 0.971 | environmental management; water environment; Huai-hai plain China; rural construction land-use change; Wuhan central China | 2011 |
| #7 | 34 | 0.899 | agent-based approach; model land-use change; farmers' decision; social-ecological framework; reforesting mountain landscape | 2006 |
| #8 | 32 | 0.996 | plant species richness; environment land use; species richness; Norwegian modern agricultural landscape; Norwegian agricultural landscape | 2003 |
| #9 | 27 | 0.981 | production-living function; spatial analysis; land use conflict potential; coastal area; southeast coast | 2017 |
| #10 | 25 | 0.986 | pollinator assemblage; changing bee; measuring natural pest suppression; different spatial scale; local variable | 2008 |
| #11 | 22 | 0.992 | European agri-environment scheme; agricultural multifunctionality; changing landscape; ecosystem service; case study | 1999 |
| #12 | 17 | 0.964 | southern Wisconsin USA; rural housing; landscape dynamics; public–private interface; tree survival | 2006 |
| #19 | 10 | 0.982 | multimodel regression approach; human driver; sociodemographic local context; recent dynamics; Mediterranean urban fringe | 2014 |
| #22 | 8 | 0.995 | landscape function; spatial characterization; terrestrial biofuel crop; ecological consideration; negotiating landscape | 2006 |
| #27 | 2 | 1 | traditional rural landscape; woodpecker; case; Transylvania Romania; conservation value | 2010 |

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
