# Peer review of "Knowledge Mapping Analysis of the Study of Rural Landscape Ecosystem Services"

_buildings, doi:10.3390/buildings12101517_

Round 1
Reviewer 1 Report
This is a bibliometric analysis of rural landscape ecosystem services which seems not to make any contribution to the existing literature or to the field.
A number of such studies, literature surveys ....etc are available at the moment.
Reviewer 2 Report
First, the conclusion is too small to elaborate clearly. The recommendations relate to the data analysis section above.
Second, the English writing level needs to be improved. In Table 4, the first letter of the country is capitalized to maintain the same format as the full text.
Thirdly, the pictures in the paper need to provide the original data information of Citespace.I need to check the accuracy of the data.
Fourth, the English level of the paper needs to be improved.
Author Response
Please see the attachment.
And the original data is located at the supplementary material.

Round 2
Reviewer 1 Report
It has been improved significantly. However, adding a section to discuss the policy issues with currently available knowledge will add some value and enhance its readability.
We can accept this version
Reviewer 2 Report
The first,in figure 6 and figure 10, Prunning:Pathfider and figure 12Prunning: None are shown in the information in the upper left corner. Please check whether the data has clips. If Citespace uses different algorithms, specify them in the main text.
Second, in 2.1, the data collection time is May 20, 2022. The time shown in Figure 6, Figure 9, Figure 10 and Figure 12 is inconsistent. Please check.
